# How Attitude and Para-Social Interaction Influence Purchase Intentions of Mukbang Users: A Mixed-Method Study [note 1]

**DOI:** 10.3390/bs13030214

**Published:** 2023-03-02

**Authors:** Hyo Geun Song, Yen-Soon Kim, Eunmin Hwang

**Affiliations:** 1William F. Harrah College of Hospitality, University of Nevada, Las Vegas, NV 89154, USA; 2Department of Hospitality and Tourism Management, University of South Alabama, Mobile, AL 36688, USA

**Keywords:** mukbang, user-generated content, purchase intention, attitude, para-social relationship, S-O-R framework, vicarious satisfaction

## Abstract

Mukbang is widely recognized as a new type of food video on user-generated content (UGC) platforms. The purpose of this study was to identify motivators to watch mukbangs and to examine the relationship between these motivators and the intention to watch mukbangs via attitudes toward mukbangs and para-social relationships. In addition, this study examined how the intention to watch mukbangs affected purchase intention. Interviews were conducted to determine the motivation factors for watching mukbangs by collecting data from mukbang viewers. The results of the interviews suggested that vicarious satisfaction, enjoyment, information, exposure, and attractiveness were motivators for watching mukbangs. Using a survey, this study collected data from 399 participants who watched mukbangs to test relationships. Using SmartPLS, structural equation modeling (SEM) was conducted. The outcomes of the SEM indicated that vicarious satisfaction, enjoyment, and information influenced the intention to watch mukbangs via attitudes toward mukbangs. The results also indicated that exposure and attractiveness had an impact on the intention to watch mukbangs via para-social relationships. Furthermore, the intention to watch mukbangs influenced the intention to purchase food items portrayed in the mukbang content. This study contributes to the literature by empirically confirming the effect of watching mukbang on purchase intention.

## 1. Introduction

Due to the spread of COVID-19, telecommuting, and quarantine periods, the usage time of YouTube has increased [1]. Moreover, the rapid adoption of the internet has facilitated the development of user-generated content (UGC) platforms. For example, YouTube plays a critical role in the production and consumption of information about food and beverages (F&B) [2]. Many people want to watch F&B reviews; mukbangs play a central role in this regard, while also providing other benefits for audiences. The most familiar and effective sensory input for people is visual; therefore, showing content via video format makes it comprehensive, entertaining, and interesting [3]. Videos can deliver information more quickly than text, which means that they have a stronger appeal to audiences.

Mukbang is a term that means “livestream eating”. This term is used to refer to videos showing an individual in the process of eating, in conjunction with other activities, such as talking or commenting on the food [4]. Mukbangs are used not only by individual users but also by companies seeking to promote and market their products [5]. The increase in mukbang viewing indicates that users enjoy the fun content of mukbangs while also seeking F&B-related information [6]. UGC platforms provide numerous opportunities for users to explore and share information about food, playing a role in connecting people in the United States with the rest of the world in real time. Considering that mukbang content records millions of views on YouTube and is considered a trend, there may be more opportunities for F&B companies to produce and provide information using this approach in the future. This indicates the need for systematic research on mukbang and the related opportunities.

Despite the popularity of video-based UGC, and mukbangs in particular, empirical studies on this phenomenon are still rare. There is an emerging body of research on the use of mukbangs to convey F&B information [7,8,9], but studies that empirically verify how video-type UGC affects actual purchasing behavior are scarce. Influencers on UGC platforms, such as YouTubers, promote not only food-related content on their channels but also develop a relationship with their audience. The size of the audience of some influencers could be comparable to traditional media, and UGC content consumption among audiences continues to rise [2]. For this reason, both the content and the content providers, as well as their use in marketing, need to be considered.

Therefore, this study focuses on mukbang content and its content providers. Following prior research and in line with the uses and gratification theory (U&GT), this study incorporates the antecedents of para-social relationships and the motivation to watch media into its research model. A para-social relationship refers to the sort of psychological interaction experienced between the viewer, the content provider, and the content [10]. Based on the literature, this study builds a research model to investigate how para-social relationships with mukbang content providers are related to the intention to watch mukbangs.

On UGC platforms such as YouTube, consumers may have both attitudes toward content and para-social relationships, which may affect their intention to watch content. Generally, users focus on content on a UGC platform, while others seek social relationships with the content providers, which means that both should be considered in the assessment of mukbangs. Studies that empirically verify how video-type UGC, such as mukbangs, affects purchasing behavior responses are scarce.

This study used the stimulus–organism–response (S-O-R) framework as an overarching theory. Prior research confirms that this framework has predictive power in how consumers react to media stimuli [11,12]. Using the S-O-R framework, the authors of [13] designated social media interactivity as the stimulus, perceived value and immersive experience as the organism, and continuous purchase intention as the response. The authors of [14] explained the formation mechanism of media users’ impulsive purchases by applying media multi-tasking as the stimulus and perceived information utility and perceived social presence as the organism. Based on the S-O-R framework, the present study examined how the motivation to watch mukbangs (stimulus) affects attitudes toward content watching and para-social relationships (organism), eventually resulting in the intention to watch content and purchase intention (response). The study used a mixed-method approach using a qualitative study (interviews) and a quantitative method (survey).

Although numerous studies have been conducted on UGC in relation to media selection, there are significant gaps in previous studies that have focused on video-based UGC content, such as mukbangs. Various studies based on media watching have not validated the relationship between media watching and item purchasing. This study extends the existing theory by making a connection between media and item purchasing. The study contributes to the literature by empirically demonstrating that media consumption leads to the real purchasing of food items. The research questions of this study are as follows.

Which factors influence the tendency toward para-social relationships and attitudes toward mukbang content on streaming platforms?

How do attitudes toward mukbang content and para-social relationships affect the intention to watch mukbangs?

How does the intention to watch mukbangs affect purchase intention?

These questions address several research gaps. Previous studies on social media have mainly focused on variables related to viewer satisfaction or information adoption [13,14]. The present study explains user behavior based on para-social relationships, attitude, and purchasing intention, factors that were not considered in previous studies. Moreover, prior studies have mainly observed behavior, using satisfaction with content or continuance use intention as the determinants [14,15]. This study comprehensively deals with social media and consumer behavior by introducing purchasing intention in addition to watching intention. Furthermore, most prior studies have been solely quantitative. The present study bridges this research gap by conducting qualitative research to establish the validity of the explanatory variables in the quantitative analysis model.

This rest of the study is divided into the following sections: Section 2 reviews the overarching theoretical framework. Section 3 describes the methodology and design of the research. Section 4 covers the analysis of the results obtained from the qualitative research. Section 5 describes the hypothesis testing process and the analysis of the results obtained from the quantitative research. Section 6 discusses the findings from Section 4 and Section 5. Section 7 presents the practical and theoretical implications of the research, describes the study limitations, and provides information about future research directions.

## 2. Literature Review

### 2.1. User-Generated Content

User-generated content refers to content created by users, which is widely considered a fundamental feature of self-created content [16]. UGC is content produced by ordinary people who do not necessarily belong to online organizations that publish professional content. With the spread of high-speed internet networks and the popularization of social networking services, producing and enjoying UGC has become a daily activity for many internet users [17]. The format of information production has also become more diverse, and includes content for passive consumption.

Research on UGC emerged alongside the development of social media, which is characterized by participation, interoperability, sociality, and convergence. This resulted in the creation and diffusion of a broad array of media content created by network users [18]. Studies on UGC have adopted diverse methodological approaches, including case studies [19,20], and have examined different platforms, including photo- and video-sharing platforms [20]. A study of a user-generated video-sharing content platform [21] confirmed the effect of users’ UGC video-viewing motivation on the functional and emotional value they gained. Ray [22] used the U&GT to identify the reason why people used food delivery apps.

As can be seen from these previous studies, research on factors impacting the motivation to watch mukbang content is scarce. Therefore, in this study, interviews were conducted with mukbang users to identify their motivations for viewing mukbang content. Motivation factors were established based on the results of these interviews as well as previous studies. In addition, this study empirically analyzes the effect of media consumption on purchasing real food items by users after watching mukbang content on UGC platforms.

### 2.2. Uses and Gratification Theory

The choice of media is important, and UGC provides access to many types of information. New types of media, such as UGC, allow people share information with others via text and video [23]. The U&GT has been applied in research on media selection to fulfill unsatisfied needs.

Media consumers’ preferences are influenced by many factors [24,25,26]. According to the U&GT, motivation and needs are commonly indicated as affecting consumer media preferences. The U&GT has also been used to study and compare passive and active users [27]. Many studies have attempted to understand how individuals make selections among all media sources. Finn & Gorr [28] examined motivations for the consumption of media content. These motivations included companionship, arousal, content, relaxation, information, escape, entertainment, passing time, and social interaction. In [29]’s research, the following motivations were established—habit, escape, arousal, passing time, relaxation, entertainment, companionship, information acquisition, and social and psychological compensation.

Recent studies on the use of new media, such as the internet and mobile devices, adopted the U&GT to understand patterns in media use and users’ needs [29]. Lim & Kumar [29] identified societal pressure, delivery experience, ease of use, quality control, customer experience, convenience, listing, and searching for restaurants as the motivations to use an app. Of these, customer experience, ease of use, listing, and searching for restaurants were found to influence the intention to use the food delivery app in question.

This discussion indicates that studies on people’s motivation to view mukbang videos are scarce. Therefore, the present study conducted interviews with mukbang users to determine their motivations, with reference to the existing body of research. Based on the U&GT, this study determined attitudes toward mukbang content to confirm the relationship between motivation and attitude. This was achieved by measuring the degree of gratification that mukbang users experience after consuming mukbang content.

### 2.3. Para-Social Relationships

One of the primary objectives of this study was to assess the impact of the para-social relationships between mukbang users and content providers. Para-social interactions comprise interactions with a person’s media personality; para-social interactions are perceived to be intimate and real due to the speaker’s body language [30]. Horton [30] defined para-social interaction as a “simulacrum of conversational give-and-take” (p. 215) that individuals experience as a response to content providers in media [31]. In the mukbang experience, for instance, viewers may be repetitively exposed to mukbang content, thereby building para-social relationships with the media performers. This is initiated by the media performer, and the relationships are not mutual [30]. Moreover, the interaction is directed by the content provider and is therefore asymmetrical [23].

Such relationships can be developed between mukbang content providers and their viewers, for instance, on YouTube. UGC users can subscribe to the content providers’ channels on the UGC platform to watch videos regularly. The development of relationships between content providers and users on UGC platforms can be encouraged by para-social relationships [30]. UGC content providers regularly address UGC users by “looking” at them through the screen [23].

Para-social interaction is associated with future media use [32]. Mukbang viewers who feel an emotional relationship with the content provider are more likely to support the content provider. Eating is not only a matter of nutritional intake but also a social behavior [33]. Viewers watch mukbangs for social reasons, such as self-presentation, belongingness, or social gratification [34]. In other words, mukbangs impart social and emotional meaning for content users via technology.

In marketing, para-social relationships play a vital role. Previous studies have observed that endorsement by a celebrity creates idealization [35]. Para-social relationships with media characters have a significant influence on the behavior of the viewer [36]. Sokolova [23] examined para-social relationships on YouTube and found that the exposure, credibility, and attractiveness of content providers has an impact on viewers’ para-social relationships. In addition, Kurtin [37] investigated the factors of para-social relationships on YouTube and found that three types of attractiveness and exposure have an effect on para-social relationships. Purwanto [38] analyzed the effect of para-social relationships on celebrities. The study showed that para-social interactions on social media have an impact on purchase intention.

### 2.4. Stimulus–Organism–Response (S-O-R) Framework

According to the S-O-R framework, stimulus cues can evoke an individual’s (organism’s) internal reaction, resulting in certain behaviors (responses) [12]. This study selected the S-O-R framework as the theoretical framework because it has been extensively confirmed in several online purchasing contexts [39,40].

Prior research has applied the S-O-R framework to purchasing behavior. Parboteeah [41] confirmed that the S-O-R framework was effective in online commerce. The study found that situational cues (S) have a significant impact on perceived enjoyment and usefulness (O). Perceived enjoyment and usefulness have an effect on the urge to make impulsive online purchases (R). Based on the S-O-R framework, Chang [11] identified relationships between website cues, traits, and the urge to buy impulsively. Moreover, the study observed that the visual appeal of a website (S) had an impact on personality (O), which, in turn, had an effect on the urge to buy impulsively (R).

Overall, prior research relating to online purchasing has examined different stimuli (e.g., situational cues, visual appeal) and internal reactions (e.g., perceived enjoyment, impulsiveness, instant gratification). Many researchers have considered purchasing intention or purchasing behavior to be a response [42]. Mukbang viewers experience vicarious satisfaction and enjoyment, as well as receiving information while they watch mukbang content. Moreover, content providers are constantly exposed to viewers, and viewers sense the content providers’ attractiveness. These factors may stimulate mukbang viewers to form relationships with content providers. Stimulated viewers may form para-social relationships with content providers. Ultimately, mukbang viewers’ intention to watch content and purchase intention will increase. Accordingly, in its contextual model, this study considered mukbang-watching motivation as the stimulus, assigned attitude and para-social relationships to an organism, and identified intention to watch mukbang and purchase intention as a response.

### 2.5. Purchase Intention and Purchase Behavior

Petcharat & Leelasantitham [43] examined individuals’ relationship with consistent online platforms and the effect of this relationship on purchase and re-purchase behaviors. The study concluded by confirming the relationship between purchase intention and purchasing behavior. Puengwattanapong & Leelasantitham [44] examined consumption behaviors by integrating the Push-Pull Mooring (PPM) model and the three stages of consumption behaviors in the online environment. The study confirmed the relationship between pre-purchase and purchase. Huang & Benyoucef [45] investigated the effects of usability, functionality, and sociability on the purchase decision-making process. The study result, namely, that purchase intention has a significant effect on purchase behavior, has been proven in several other studies. Therefore, the present study aims to investigate purchasing behavior by measuring purchase intention.

## 3. Research Methodology

In this mixed-method study, the two-step approach suggested by [46] is adopted, with a qualitative approach used in Phase 1 and a quantitative approach used in Phase 2. The central goal of this study is to explain the motivations behind mukbang watching. Following a mixed-method research design as suggested by [39], motivation factors were first developed from interviews conducted in Phase 1. Theories were then adopted to explore mukbang watching. In Phase 2, a survey was conducted to gain further empirical tests for the theoretical framework in this study.

A phenomenology approach that involved interpretative phenomenological analysis (IPA) was utilized to complete Phase 1. According to [47], this approach focuses on understanding the participants’ experiences as a product of their interactions with their environment. In such cases, the researcher uses an interview to explore personal insights and perspectives of individuals who have watched a mukbang video. The goal of the present study is to gain an in-depth understanding of the mukbang follower; therefore, result, a qualitative approach was adopted to gain unquantified insights into the participants’ own experiences.

In Phase 2 of the quantitative study, the research model and hypotheses were constructed based on previous research. Then, samples were collected via a survey questionnaire. This study analyzed the hypotheses using partial least squares (PLS). Venkatesh et al. [46] discussed the key elements of mixed-method research, indicating that findings from the qualitative approach should be integrated first, and from the quantitative approach last.

## 4. Qualitative Design and Data Analysis (Phase 1)

Determining the reasons for viewing is difficult, as watching eating and drinking online content is a new trend, and there are few related studies. For this reason, an in-depth explorative analysis using interviews was necessary. This study used interviews to examine users who watched mukbangs on UGC platforms. Using the U&GT, which is a critical approach for studying new media, this study examines the attributes of the motivations for watching mukbangs using the interview method. The main objective of qualitative research is to gain an in-depth understanding of a phenomenon [48]. Hence, most qualitative research uses non-random sampling [49]. In the present study, the researcher used interviews to obtain the personal insights and perspectives of individuals who had watched mukbangs.

A phenomenology approach was utilized in Phase 1, namely, interpretative phenomenological analysis (IPA). According to [47], this approach focuses on questions about participants’ experiences and the relationship between the subject and world. According to [48], phenomenology is a genre of quantitative research that is used to describe the meaning of people’s lived experiences of a particular phenomenon. The present study explored the meaning of the lived experiences of mukbang viewers. Hence, qualitative research methods were deemed appropriate for this study, and a phenomenological approach was used to understand the new phenomenon of mukbang videos. Moreover, this study used interactive and humanistic methods.

### 4.1. Qualitative Data Collection

The number of interview participants in this study was 15, consisting of 8 men and 7 women. The sample was recruited using online recruitment in the Las Vegas area in United States, and not only at the University of Nevada, Las Vegas. This method was not only convenient for researchers, but was also a good method of achieving greater generalizability. The aim of this study was to explore the views of individuals who have watched mukbang videos and to determine how they have been affected by their experiences. Therefore, all participants had viewed mukbangs. These individuals grew up with technology, and rely on and enjoy online platforms such as YouTube or Twitch. They enjoy communicating via new social media platforms and sharing what they enjoy, such as mukbang videos. The interview data collection was conducted from 27 to 28 September 2020.

Qualitative research consisted of semi-structured interviews, moderated by the researchers first preparing interview guidelines to ask the respondents about their perspectives. These guidelines allowed for flexibility to change the direction of the questions during the interviews. Qualitative research comprises semi-structured interviews moderated by researchers by preparing guidelines for the interviews, asking the respondents about their perspectives, and then flexibly changing the direction and questions according to the flow of the interview. The questionnaires comprised three sections—(1) reasons for watching mukbangs, (2) perspectives on the mukbang content providers, and (3) detailed descriptions of participants’ experience of mukbang watching. This study conducted a total of 15 personalized in-depth interviews. The participants reviewed the definitions of terms regarding mukbangs and the descriptions of a mukbang-watching experience example before the main interview to understand contextual information. The time for the interviews ranged from 15 to 40 min. With the respondents’ consent, all interviews were digitally recorded.

### 4.2. Qualitative Data Analysis

The qualitative research results on the motivation for watching mukbang videos were summarized by integrating the content of the interviews using open coding. This was used to categorize the data into concepts. This categorization was derived from “means phenomena” and data [50]. Naming the categories and codes resulted in clear groupings of content under broad descriptions; data were described clearly and the material obtained from the data was recalled [51]. The word coding of all interview materials was conducted using MAXQDA software. Open coding involved a line-by-line analysis of sentences, expressions, and words. Conceptualization was only included if the analysis revealed similar content without considering the phrases, words, and clauses of the collected data or repetition. After conducting the open coding, the concepts were minimized by using axial coding. This resulted in the included content appearing within several contexts and in the responses of the research participants. Thus, 27 categories were derived from 60 concepts. These categories were then divided into five core categories using selective coding.

These were grouped into five categories of similar concepts, in line with previous research (see Table 1). These were (i) vicarious satisfaction, (ii) information, (iii) enjoyment, (iv) exposure, and (v) attractiveness. The variables (motivations to watch mukbangs) were determined based on the results of the interviews. Numerous existing studies related to TV-watching motivations were compared with the results of the interviews, since viewing mukbangs involves watching and consuming media. Many variables related to TV-watching motivations have been developed and used in prior studies, and their reliability and validity have been confirmed. It was noted that motivation factors were complex, and no single factor simultaneously explained the viewing behavior of an interviewee. The motivations identified in the current study had the same meanings as those developed in previous studies, and these could be applied and slightly modified as motivation factors for watching mukbangs.

## 5. Quantitative Design and Data Analysis (Phase 2)

### 5.1. Development of Research Models Based on the Literature and Qualitative Study

In Phase 2, the conceptual research model was constructed based on prior qualitative studies and literature reviews. According to the U&GT, both active and passive audiences choose media to obtain gratification [32]. However, there are differences between the two groups. Passive audiences absorb media content through media performers, while active audiences actively choose media content based on their own beliefs [58].

Following prior research and the U&GT, this study incorporates the antecedents of motivation and para-social relationships into the model. Based on social cognitive theory, this study focuses on the effect of para-social relationships on the intention to watch mukbangs. The study also focuses on attitudes toward mukbangs and their effect on the intention to watch mukbangs, as well as attitudes toward food. The coding of the qualitative research revealed comments confirming that vicarious satisfaction, enjoyment, and information affect attitude. We also found repeated statements that exposure and attractiveness are related to para-social relationships. Based on this theoretical foundation, the following research model is proposed (see Figure 1).

### 5.2. Hypotheses Development

One of the main characteristics of the U&GT is the premise that users choose media based on unsatisfied needs. Thus, users’ choices stem from their motivation [23]. Users have various motivations to consume the same content to fulfill unsatisfied needs. Media consumers continue to use the media when they believe that their needs will be met. The stronger the relationship between motivation and attitude, the more satisfied the user, which leads to increased use of media [24].

Motivation has been identified as a predictor of attitudes in U&GT studies [59]. Mehrad & Tajer [60] examined how motivation to use media affected social media use, concluding that motivations for social media use have a positive effect on gratification. Kim [6] proposed a relationship between motivation and gratification in a study examining an evening news program. The study suggested that the motivation to view news programs has an impact on gratification derived from content.

Research on the motivation to watch UGC [61] identified that information and enjoyment are a vital motivation for the use of UGC on social media. Gogan [15] confirmed that entertainment, social interaction, and information have a positive impact on satisfaction obtained from media use. Hossain [62] verified that enjoyment and information play a vital role in social media use. Furthermore, the vicarious satisfaction derived from the mukbang content creator is adapted to mukbang viewers [63], which is a unique characteristic of mukbangs. According to [64], vicarious satisfaction is a major reason for viewing mukbang videos. A recent study claimed that individuals fulfill their need to eat in the company of others by feeling emotionally connected with the mukbang content providers and other viewers [5]. Kircaburun et al. [34] argues that watching mukbang videos helps individuals overcome loneliness and alienation by providing a sense of community.

Mukbangs are a source of information about food that many people like to eat. It is easy to filter and watch information on food ingredients or products, and use this to appraise information based on the reviews of users who have already watched the content. This suggests that the stronger a consumer’s desire to watch mukbang videos, the more the content will satisfy their needs. Based on previous studies, it can be hypothesized that the motivation to watch mukbang videos positively influences attitudes toward mukbangs.

Based on qualitative research and prior research, this study supposes that vicarious satisfaction, enjoyment, and information have a positive impact on attitudes toward mukbangs on UGC platforms. This would affect attitudes toward food and intention to watch mukbang videos created by viewers’ favorite content providers, and indirectly have an impact on purchase intention.

**H1.** 
*Vicarious satisfaction is positively related to attitudes toward mukbangs.*


**H2.** 
*Enjoyment is positively related to attitudes toward mukbangs.*


**H3.** 
*Information is positively related to attitudes toward mukbangs.*


Para-social relationships with attractive characters have been found to be important in media use [30,38]. Online celebrities are influential, as viewers can easily have relationships with them by posting comments or through donations [23]. According to prior research on para-social relationships, repeated viewer exposure to content providers can help build and maintain para-social relationships [31]. Sokolova [23] confirmed that exposure influences para-social relationships. Moreover, the content providers’ attractiveness plays an important role in para-social relationships [10]. In other words, a user who finds a content provider attractive is more likely to have a para-social relationship with that content provider than another user [32]. Therefore, the following hypotheses were tested in this study:

**H4.** 
*Exposure is positively related to para-social relationships.*


**H5.** 
*Attractiveness is positively related to para-social relationships.*


Attitude refers to a lasting general assessment of people, objects, advertisements, or issues [65]. It is the individual’s general evaluation of a concept that comprises the overall emotion of favorability [66]. Overall, the evaluation is based on necessary and detailed information [67]. Internet users can find precise product information in an online environment [68]. The UGC is a useful tool for providing detailed information by presenting vivid product reviews [2]. Online consumers prefer the visual demonstration of items, which makes those products appealing [69]. The relationship between attitude and intention is an extended application of the TPB [70,71]. Previous studies using this theory have demonstrated many of these connections, especially in multimedia studies. Therefore, this study assumes that the attitude toward mukbang videos is related to the intention to watch mukbang videos.

**H6.** 
*Attitudes toward mukbang*
*content are positively related to the*
*intention to watch mukbangs.*


Para-social relationships are illusory relationships established between a viewer and a performer. These relationships can be developed between YouTubers and their followers [72]. Although such relationships are primarily unidirectional, the possibility of liking and commenting could give the impression that there is a communication process [73]. According to [32], future media use is influenced by para-social interactions. Najar [73] showed that social action is one of the key factors driving media watching in the hospitality and tourism fields. Viewers who form a para-social relationship are more likely to want to watch the content provider’s videos than those who do not form such a relationship [72]. Thus, this study proposes that para-social relationships are related to the intention to watch mukbang videos.

**H7.** 
*Para-social relationships are positively related to the intention to watch mukbangs.*


Prior research has shown that attitudes toward a service or product affect the intention to provide services [74]. According to the theory of planned behavior, attitudes toward intention explain intention [75]. Castaneda [76] confirmed that attitudes toward a website influence the intention to visit the website. Numerous studies on information systems have examined the role of attitudes toward content in predicting the intention to watch content [39,77]. According to [73], the intention to watch UGC content has a positive impact on the intention to purchase food items in restaurants. Therefore, this study tested the following hypothesis:

**H8.** 
*Intention to watch mukbang is positively related to purchase intention.*


### 5.3. Ethical Considerations

The Institutional Review Board (IRB) at the University of Nevada, Las Vegas approved this study. Before conducting the interviews and surveys, the participants were informed that participation was voluntary, all personal profiles would be kept confidential, and data would only be used for the purposes of the study. Ethical guidelines related to conducting surveys and interviews were adhered to. The inclusion of private information, such as names and job titles, was avoided because most respondents would have been reluctant to take part in the research otherwise. Maximum efforts were made to ensure anonymity so that respondents could not be identified. In the context of research, ethics refers to the appropriateness of behavior in relation to the rights of those who are the subjects of the research or are affected by it. The importance of anonymity was explained to the participants taking part in the qualitative and quantitative studies, as well as the overall significance of the study, to ensure the participants’ commitment to the research.

### 5.4. Measurement

The measurement methods used in this study were taken from prior studies to ensure reliability and validity, but they were slightly modified. Based on [55]’s research, vicarious gratification is defined as “the perceived feeling experienced by watching other people eating in mukbang, rather than by eating food by oneself.” Exposure is defined as “the extent to which mukbang watchers perceive that they watch mukbang without special purpose”. Enjoyment is defined as “a perception of the fun and enjoy activity derived from watching mukbang” [54]. Information is defined as “the extent to which mukbang contents shared in UGC can provide users with relevant and timely information” [70]. Attractiveness is defined as “the extent to which mukbang users consider mukbang content providers’ charming” [74]. Purchase intention is the “possibility that mukbang users will purchase the products shown in mukbang content” [57].

All measurement items were measured on a seven-point Likert scale ranging from 1 (strongly disagree) to 7 (strongly agree). A total of thirty measurement items was used, as shown in Table 1, namely, vicarious gratification (three items), enjoyment (three items), information (four items), exposure (three items), attractiveness (three items), attitude toward mukbang (four items), para-social relationships (three items), intention to watch mukbang (four items), and purchase intention (three items).

### 5.5. Quantitative Data Collection

For the quantitative research, the sample consisted of respondents who had watched mukbang content in the past 12 months. The aim was to explore their opinions and experiences. This study asked participants to recall their last mukbang watching experience. Qualtrics, a professional online research firm, performed the survey via web-link. Qualtrics recruited the respondents and administered the questionnaire from 7 to 10 October 2020 to mukbang content users in the United States. This included a consultation with the research designers from the company.

The sample size was 399 and represented a variety of backgrounds. Approximately 65.1% of the respondents were male, and 34.9% were female. In total, 42.1% of respondents were aged between 20 and 29 years. Other groups represented were those aged between 30 and 39 years (27.8%), 40 and 49 years (16.5%), and 50 years and above (10.0%). In terms of annual house income levels, 22.1% earned USD 50,000 or less, 47.6% earned between USD 50,001 and USD 150,000, and 30.3% earned more than USD 150,000.

### 5.6. Data Analysis

Data screening was preceded by pre-analysis to validate the importance of the data. According to [78], pre-analysis data screening detects any irregularities or problems with the collected data. Faubion [79] indicates that pre-analysis is vital to ensuring the accuracy of data. The tool used to collect data in this study was a web survey, which helped to eliminate inaccurate data. Pre-data analysis and screening detected and eliminated incorrect responses [78]. The study adopted the suggestion of [80] to address data screening, which involved conducting a visual inspection while collecting data to eliminate any responses that completely resembled each and that could therefore be biased. The downside of the method is potentially losing the partially collected data needed for the study because the data are essential for data screening. However, losing partial data was necessary to increase data validity. Hair et al. [81] noted that using data without pre-analysis can lead to false results. Data pre-analysis and screening help to identify multivariate outliers, which can then be eliminated.

By conducting data screening, the validity of the research model was confirmed. The proposed research model was tested using partial least squares structural equation modeling (PLS-SEM). PLS-SEM is a useful tool for assessing multiple and complex relationships among variables [82], such as those in this study. This study also sought to evaluate causal relationships [83], and PLS-SEM is an appropriate tool for causal analysis. A two-step process was followed—measurement model analysis and structural equation modeling [81]. SmartPLS 3 was used for this analysis.

#### 5.6.1. Measurement Model Assessment

Harman’s single-factor test assesses common method variance [84]. The test yielded nine factors with eigenvalues greater than 1. The largest factor accounted for 35%, confirming the lack of common method variance.

After convergent validity was assessed by the significance of the loadings and the average amount of variance extracted (AVE greater than 0.5) [85,86], one item was eliminated from further analysis because it had a low loading value. The remaining loading values of all items were greater than 0.7. All AVEs exceeded 0.5 [86]. All Cronbach’s alpha values exceeded 0.7, establishing reliability and convergent validity [87]. Construct reliability was assessed via internal consistency and indicator reliability [81].

This study confirmed the items’ cross-loadings and observed that all items had high loadings values on their corresponding constructs [81]. To evaluate discriminant validity, this study evaluated whether the square root of the AVE of the constructs was greater than the correlations between that construct and the other constructs. Table 2 shows that all factors achieved this condition [86]. Overall, the tests demonstrated that each measurement item was valid and reliable.

The overall model fit was measured via four parameters. Thus, the study measured d_ULS, d_G, the standardized root mean square residual (SRMR), and the normed fit index (NFI) [88]. By using d_ULS and d_G, the exact model fit-tests the statistical inference of the inconsistency between the covariance matrix. This is inferred from the composite factor model and the empirical covariance matrix. The values of d_ULS and d_G should be lower than the 0.99, which indicates a good measurement model fit. The SRMR refer to the difference between the observed correlations, and should be below 0.08 [89]. The NFI measures the Chi-square score of the suggested model and compares this with a significant benchmark [90]. The NFI has to be larger than 0.90 [90]. The d_ULS was 0.894 and d_G was 0.631, suggesting an acceptable level. The SRMR value was 0.044, indicating the data fitted the model well. The NFI was 0.865, slightly below the required criteria. However, since three indicators showed satisfactory results, this study proceeded to conduct structural model assessment.

#### 5.6.2. Structural Model Assessment

Based on bootstrapping with 1000 subsamples, the t-values and the associated *p*-values were calculated to test the hypotheses. Figure 2 and Table 3 show the outcomes of the analysis. All paths were supported at α = 0.05. Vicarious satisfaction (β = 0.526, t = 10.203, *p* < 0.001) was found to have a significant effect on attitudes toward mukbangs, while enjoyment (β = 0.218, t = 5.139, *p* < 0.001) and information (β = 0.175, t = 3.702, *p* < 0.001) explained attitudes toward mukbangs. Both exposure (β = 0.390, t = 5.559, *p* < 0.001) and attractiveness (β = 0.198, t = 2.716, *p* < 0.01) explained the para-social relationships. Hence, H1, H2, H3, H4, and H5 were confirmed. Attitudes toward mukbangs (β = 0.659, t = 13.837, *p* < 0.001) and para-social relationships (β = 0.210, t = 4.082, *p* < 0.001) were found to have a positive effect on the intention to watch mukbangs. The intention to watch mukbangs (β = 0.686, t = 15.436, *p* < 0.001) significantly explained purchase intention. Thus, H6, H7, and H8 were confirmed. The squared multiple correlations (SMCs; R^2^) for the structural equations for attitudes toward mukbangs, para-social relationships, intention to watch mukbangs, and purchase intention were 0.591, 0.285, 0.618, and 0.469, respectively.

The study tested the mediation effect of attitude and para-social relationships on the relationship between the 5 motivations and the effect of this relationship on intention to watch mukbangs. The results of this test also provided data on specific indirect effects. This study found that 5 specific indirect effects were positive in supporting the mediation effects of para-social relationships and attitude in the relationships between vicarious satisfaction and intention to watch mukbangs (β = 0.347, t-value = 7.422, *p* < 0.000); enjoyment and intention to watch mukbangs (β = 0.144, t-value = 5.162, *p* < 0.000); information and intention to watch mukbangs (β = 0.115, t-value = 3.442, *p* < 0.001); exposure and intention to watch mukbangs (β = 0.082, t-value = 2.667, *p* < 0.008); and attractiveness and intention to watch mukbangs (β = 0.042, t-value = 2.149, *p* < 0.032).

## 6. Discussion

### 6.1. Qualitative Research

The qualitative analysis results of Phase 1 revealed the five motivations for watching mukbangs: vicarious satisfaction, information, enjoyment, exposure, and attractiveness. Vicarious satisfaction was found to be a motivation for watching mukbangs. These results are similar to research on watching media [91]. This may be because the appetite is sated by proxy when viewing someone else eat and taste delicious dishes, and by the sight of food being eaten. The results showed that one major reason the participants watched the mukbang videos was because of their interesting content. This result supports prior studies on TV watching [92]. According to these studies, individuals tend not to watch media, no matter how useful it is, unless it is not entertaining. Hence, enjoyment is a crucial motivation for watching media such as mukbangs. Moreover, this study found that information was one of the motivations for watching mukbangs. The mukbang content provider created information on the food items as well as providing their views on the food and their preferences. The qualitative findings of this study show that enjoyment is an important motivator for watching mukbang content. In addition, the exposure of content providers is another motivation for watching mukbang content. This is because individuals tend to habitually seek exposure to their favorite content providers rather than seeking out simple media content, such as mukbang, from time to time [64]. The study results also revealed that the attractiveness of content providers is one of the reasons why people watch mukbang videos. Pereira et al. [56] confirmed that viewers admire the content providers’ attractiveness, which is one of the vital reasons for watching media.

The qualitative results of this study addressed both content-related factors (e.g., vicarious satisfaction, information, and enjoyment) and content provider-related factors (e.g., exposure and attractiveness). The qualitative findings of Phase I were confirmed by quantitative data analyses, whereby vicarious satisfaction, information, and enjoyment had a significant effect on attitude toward mukbangs. Furthermore, exposure and attractiveness had an impact on para-social interactions with mukbang content providers. The qualitative findings of Phase 1 can be generalized via the quantitative research of this study, with the mixed-method approach of this study successfully bridging the gap between qualitative and quantitative research. Furthermore, this link exploits the advantages of both research methods. Thus, the mixed-method approach of this study offers a deeper understanding of motivations to watch mukbangs than a single research approach.

### 6.2. Quantitative Research

The results of the SEM revealed several interesting facts. Vicarious satisfaction was found to have an impact on attitudes toward mukbang content (H1 was supported). This result is confirmed by previous studies on TV watching [63]. This finding suggests that users who derive vicarious satisfaction from watching mukbangs develop a stronger interest in mukbang content.

Enjoyment was found to be a strong predictor of attitudes toward mukbang content (H2 was supported). This result confirms the findings of prior studies [93], in which interesting content plays a vital role in stimulating positive attitudes toward mukbang content. According to flow theory, perceived enjoyment plays a vital role, and leads to user addiction in many fields [94]. According to [5], some individuals simply seek entertainment from watching mukbang content.

In line with prior TV-related research [63], the study findings reveal that information has a significant association with attitudes toward content (H3 was supported). For example, viewers who are motivated to pursue information about media content will have a higher level of interest in mukbang content than in the objects portrayed in the program. This may be because viewers who perceive the information provided in mukbangs as useful tend to form favorable attitudes toward mukbangs.

Moreover, this study showed that exposure has an impact on para-social relationships (H4 was supported). This result also confirms prior research [90]. Considering the predictors (antecedent, driver, precursor, and enabler) of para-social relationships, repeated viewer exposure to mukbang content providers can help develop para-social relationships [93].

The attractiveness of content providers appeared to be an additional predictor of para-social relationships (H5 was supported). This outcome was also similar to the results of a previous study on YouTube [53]. The attractiveness of the speakers was related to para-social relationships with the television speakers [32]. This means that viewers who often watch the content of a mukbang content provider and who find that content provider more attractive are more likely to exhibit a stronger para-social relationship with that content provider than other viewers. Thus, this study confirmed that attractiveness has an impact on para-social relationships.

Attitudes toward mukbang content had an effect on the intention to watch mukbangs (H6 was supported). The relationship between attitude and intention has also been supported in previous research [75]. This finding suggests that viewers who have a good experience with mukbang content develop a stronger interest in watching mukbang content. Our research also confirmed that attitudes toward mukbang content influenced the intention to watch mukbangs.

The study results revealed that para-social relationships also had an impact on the intention to watch mukbangs (H7 was supported). These findings are in line with related research [23]. One possible explanation is that watching mukbang videos helped individuals overcome loneliness and alienation by providing a sense of community [94]. This study also showed that para-social relationships are linked to the intention to watch mukbang content.

The study findings revealed that attitude toward mukbang content made a stronger impact on the intention to watch mukbangs than para-social relationships. This outcome is in line with related research [72]. This indicates that future researchers who are interested in the subject of mukbangs should consider attitudes before para-social relationships.

The intention to watch mukbang content plays an important role in determining individuals’ intention to purchase food items shown in the mukbang videos (H8 was supported). Therefore, we conclude that well-developed mukbang content does indeed induce viewers to purchase real items. The link between online content and purchasing physical items has been empirically established [95]. This relationship is consistent with prior research, indicating that media content has a reinforcing effect on purchasing real items [63]. This study also demonstrated that the specific use of online content is linked with the intention to purchase a real item that was shown in online content.

## 7. Conclusions

### 7.1. Theoretical Contributions

This study makes several contributions to the body of research. First, it confirms the relationship between watching media and purchasing behavior. In particular, this study focuses on the direct relationship between the intention to watch mukbangs and the intention to purchase a food item observed in a mukbang video. This study provides valuable insights into the role of UGC content and content providers in influencing purchase intentions. Furthermore, the study sheds light on consumers’ purchasing behavior and post-purchase behavior by examining the intention to watch content and purchasing intention. In this study, the intention to watch content and purchasing intention correspond to the stages of the online purchase decision-making process described by [43]. According to [43], intention recognition enhances purchase behavior, eventually leading the individual to recommend and re-purchase. Therefore, individuals with a high viewing intention and a high purchasing intention may actually purchase services or products. Scholars can empirically analyze actual purchase behavior and post-purchase behavior in the context of this study. In addition, the watching intention and purchase intention in this study correspond to the pre-purchase process of the consumer behavior stages described by [44]. Puengwattanapong & Leelasantitham [44] confirmed that pre-purchase behavior improves purchase, resulting in post-purchase. To sum up, users with a high intention to watch content and make a purchase will actually make the purchase and perform post-purchase actions. Based on the results of this study, scholars can investigate the effect of watching media on purchasing behavior.

Second, this study contributes to the generalizability of the S-O-R framework [12] by validating the explanatory power of its theoretical dimensions in a new context: watching mukbangs. Thus, this study confirmed the role of the S-O-R framework, which has been robustly verified in different fields, by demonstrating its application in the context of mukbangs. Lee [96] explained that generalizability is vital in academic studies since it reaffirms a theory’s usefulness. In other words, the more a theory is confirmed in different fields, the greater its usefulness in explaining a phenomenon. Therefore, through the results of this study, future researchers will be able to expand and apply the S-O-R framework to the context of UGC.

Third, this study examined a new construct that has been bypassed by many mukbang studies, namely, vicarious satisfaction. No studies have examined this concept in relation to motivation to watch mukbangs. This study confirms a new source of motivation to watch mukbangs by demonstrating that vicarious satisfaction is a vital driver of motivation for potential mukbang viewers. Thus, the study contributes to the body of research by introducing a new concept into the research on mukbangs and UGC. Based on this study, future researchers could investigate the determinants of vicarious satisfaction. Since vicarious satisfaction can change depending on viewers’ economic situation or emotional factors, future research could conduct additional tests with several control variables, such as income, mood, or environmental status.

Finally, this study examined the motivation for watching mukbangs by using a mixed-method quantitative and qualitative approach. The closed-ended questionnaire survey did not provide an in-depth understanding of the new phenomenon [97]. However, the aim of this study was to provide a greater understanding of the motivation to watch mukbangs; therefore, a qualitative approach was adopted to gain unquantified insights into the participants’ own experiences. Thus, by combining qualitative and quantitative research, this study conducted a more accurate analysis. The empirical results of the research model make a meaningful contribution to the literature by incorporating the qualitative findings on motivation to watch mukbangs with a quantitative analysis of the relationships between key influencing factors. Using this procedure, this study provides a new and comprehensive framework for understanding the motivations behind watching mukbangs on UGC platforms.

### 7.2. Practical Implications

This study also has practical implications, which are listed below. First, by showing the relationship between watching mukbangs and purchase intention, the study indicates how the probability of watching mukbang content can lead to real purchasing behavior. When a company advertises a specific product, watching mukbangs may affect purchasing intention. In the food and beverage industry, it is possible for sellers to consider strategic sponsorship with famous mukbang content providers to increase sales. Via this strategy, mukbang content providers can talk to viewers about new food items. Moreover, companies can gain new opportunities for sales and attract new customers by promoting their products through mukbang content.

Second, this study examined whether the para-social relationships between viewers and content providers play increase the intention to watch mukbangs. The results show that para-social relationships are a vital predictor of watching mukbangs. The significant association between para-social relationships and watching UGC content is also supported by [74]. Indeed, the relationship between YouTubers and viewers has been created and strengthened in a variety of ways. Mukbang content providers talk about food items in a natural manner, and mukbang viewers accept these comments more readily. Hence, it would be beneficial for food marketing professionals to communicate with their future customers via famous mukbang content providers.

Lastly, considering the predictors of para-social relationships, our results are in line with the literature, confirming that exposure and attractiveness can help build virtual proximity with a content provider [23]. Quantitative marketers can identify viewers’ viewing patterns in detail using big data analysis. By applying these methods, they can continuously stream mukbang content during high viewing periods, thereby strengthening para-social relationships. Currently, many mukbang content providers operate channels based on their own themes and characters. If the content provider includes attractive factors that enhance the relationship with viewers and allow viewers to form a unique identity, they will be able to attract more viewers. For example, YouTubers can invite celebrities to mukbangs or plan product placement events in real time to attract more viewers. Based on these strategies, companies can increase sales.

### 7.3. Limitations and Future Research Directions

This study has three limitations. First, it does not examine different types of mukbang content. Mukbangs may differ according to the characteristics of the providers (i.e., a mukbang on cooking or ASMR). Thus, since information is more important for viewers of cooking-based mukbangs than ASMR-based mukbangs, information may play a lesser role in ASMR-based mukbangs. Furthermore, viewers generally prefer a specific type of mukbang, which may affect their intention to watch mukbangs or their purchase intention. Hence, further research is needed in order to explore several contexts. Second, although we sought to reflect the real world to the greatest extent possible in our study, a quantitative analysis of actual purchase behavior, sales increases, or profits was not performed. Therefore, future studies are needed in order to verify the quantitative and objective effects of each variable in determining user intention. Finally, this study conducted a cross-sectional analysis. However, viewers may act differently depending on the time or environmental situations. Therefore, a follow-up study using a longitudinal approach is necessary to observe the dynamic effects of watching mukbangs.

## Figures and Tables

**Figure 1 behavsci-13-00214-f001:**
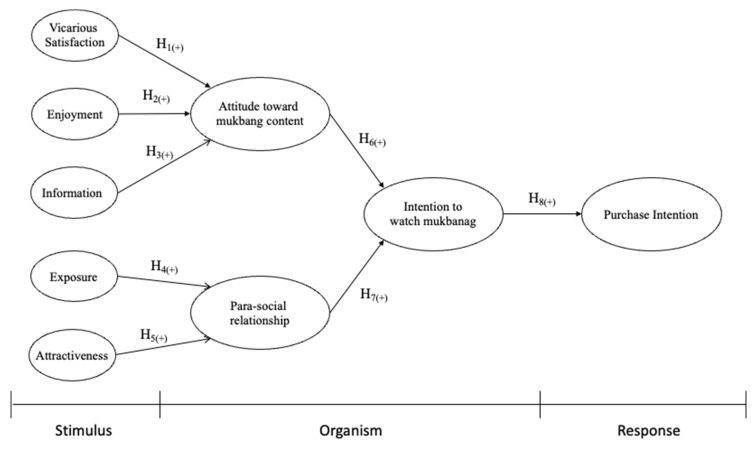
Research model.

**Figure 2 behavsci-13-00214-f002:**
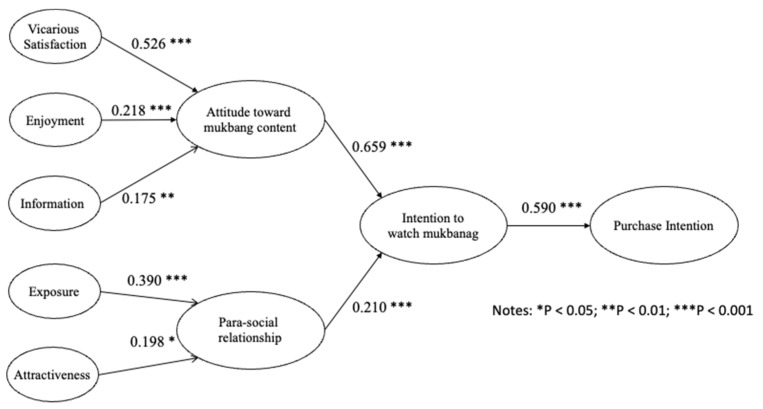
Results of the SEM.

**Table 1 behavsci-13-00214-t001:** Instrument items.

Construct	Item	Loadings	α ^(1)^	C.R ^(2)^	AVE ^(3)^	Items’ Origin
Attitude toward mukbangs	I feel good watching mukbang content.	0.877	0.919	0.943	0.805	[52]
	I like watching mukbang content on YouTube.	0.908				
	It is wise to watch mukbang content on YouTube.	0.909				
	It is worthwhile watching mukbang content on YouTube.	0.895				
Attractiveness	I find that the mukbang YouTuber is attractive.	0.915	0.815	0.890	0.729	[53]
	I think the mukbang YouTuber is quite enticing.	0.888				
	The mukbang YouTuber is charming.	0.889				
Enjoyment	I have fun with mukbang.	0.864	0.907	0.935	0.782	[54]
	Watching mukbang provides me with a lot of enjoyment.	0.836				
	I enjoy watching mukbang.	0.929				
	Watching mukbang makes me excited.	0.906				
Exposure	Mukbang is good for spending time alone.	0.915	0.880	0.926	0.806	[23]
	I watch mukbang habitually without a special purpose.	0.888				
	There are no other programs to watch at the time this program is broadcast.	0.889				
Information	Mukbang helps me to store useful information about food.	0.867	0.920	0.943	0.806	[55]
	Mukbang is good to retrieve information about food when I need it.	0.913				
	To keep up to date on the latest news and events about food, mukbang is useful.	0.919				
	Mukbang is useful for finding useful information about food.	0.892				
Intention to watch mukbangs	I intend to watch mukbang frequently.	0.924	0.919	0.949	0.860	[56]
	I plan to watch mukbang more often.	0.938				
	My willingness to watch mukbang is high.	-				
	The probability that I would consider buying an F&B product that I saw in a video is high.	0.920				
Para-social relationships	The mukbang influencer makes me feel comfortable, as if I was with a friend.	0.920	0.919	0.949	0.860	[23]
	I look forward to watching the last video uploaded by the mukbang influencer on YouTube.	0.938				
	If the mukbang influencer appeared on another form of media, I would watch it to know more.	0.924				
Purchase intention	If I were to buy an F&B product, I would consider buying what I watch on a mukbang video.	0.928	0.894	0.934	0.824	[57]
	The likelihood of me purchasing an F&B product that I saw on a mukbang video is high.	0.886				
	My willingness to buy an F&B product that I saw on a mukbang video is high.	0.909				
Vicarious satisfaction	While watching mukbangs, I can forget my daily life.	0.948	0.926	0.953	0.871	[55]
	While watching mukbangs, I feel like those who eat the food.	0.934				
	While watching mukbangs, I feel like I am eating.	0.918				

Note. ^(1)^ Cronbach’s alpha; ^(2)^ composite reliability; ^(3)^ average variance extracted.

**Table 2 behavsci-13-00214-t002:** Correlations among constructs.

	(1)	(2)	(3)	(4)	(5)	(6)	(7)	(8)	(9)
(1) Attitude toward mukbangs	0.897 **								
(2) Attractiveness	0.479	0.854 **							
(3) Enjoyment	0.563	0.561	0.885 **						
(4) Exposure	0.739	0.631	0.458	0.898 **					
(5) Information	0.548	0.503	0.552	0.478	0.898 **				
(6) Intention to watch	0.466	0.694	0.460	0.483	0.475	0.927 **			
(7) Para-social relationships	0.509	0.444	0.620	0.514	0.432	0.546	0.927 **		
(8) Purchase intention	0.732	0.609	0.545	0.660	0.401	0.685	0.517	0.908 **	
(9) Vicarious satisfaction	0.713	0.722	0.472	0.708	0.480	0.714	0.528	0.630	0.933 **

Note. ** *p* < 0.01.

**Table 3 behavsci-13-00214-t003:** Standardized structural estimates and tests of the hypotheses.

H	Path	t-Value	Estimates	Results
H1	Vicarious satisfaction	→	Attitude toward mukbangs	10.203	0.526	Supported
H2	Enjoyment	→	Attitude toward mukbangs	5.139	0.218	Supported
H3	Information	→	Attitude toward mukbangs	3.702	0.175	Supported
H4	Exposure	→	Para-social relationships	5.559	0.390	Supported
H5	Attractiveness	→	Para-social relationships	2.716	0.198	Supported
H6	Attitude toward mukbangs	→	Intention to watch mukbangs	13.837	0.659	Supported
H7	Para-social relationships	→	Intention to watch mukbangs	4.082	0.210	Supported
H8	Intention to watch mukbangs	→	Purchase intention	15.436	0.686	Supported
R^2^					
Attitude toward mukbang	0.591 (59.1%)			
Para-social relationships	0.285 (28.5%)			
Intention to watch mukbangs	0.618 (61.8%)			
Purchase intention	0.469 (46.9%)

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
