# Peer review of "How Attitude and Para-Social Interaction Influence Purchase Intentions of Mukbang Users: A Mixed-Method Study [Author-notes fn1-behavsci-13-00214]"

_behavsci, 2023, doi:10.3390/bs13030214_

Round 1
Reviewer 1 Report
This research is well structured and prepared, it is perfectly follow mixed method approach regardless insufficient details of derived themes and model.
it needs just minor proofreading and authors might reconsider introducing the title to be worldly viewed and get higher chances for generalization.
Author Response
I would like to express gratitude to Reviewer 1 for the valuable review. After revision, I realized that I are indebted to the reviewer for every improvement made in the paper. The comments from Reviewer 1 and the corresponding replies/revisions are attached in the docu file.

Reviewer 2 Report
This study was done quite well and the resulting paper was written quite well. But I have three comments which could help further improve your paper. First, you talked very little of the research gaps. You need to declare whether something similar to your study had been done before. If yes, then what is the key difference between your paper and these other similar papers. If so, then you need to declare this/these as research gap(s) and argue why it is worth researching along this/these gap(s). Second, the sources of the items you used for measuring the different variables are unclear. Please write more clearly where did these items originate. Third, you might have mentioned this and I might have overlooked this but for now I can't find out in which country the primary data for this study was collected. Please specify which country you collected primary data from.
Author Response
I would like to express gratitude to Reviewer 2 for the valuable review. After revision, I realized that I are indebted to the reviewer for every improvement made in the paper. The comments from Reviewer 2 and the corresponding replies/revisions are attached in the docu file.

Reviewer 3 Report
The research is interesting, but there are some aspects for improvement.
1. The title is quite long. It will be better to leave one main idea.
2. What is the main purpose of your research?
“The purpose of this research was to identify motivators and to examine the relationship between motivators and the intention to watch mukbang via attitudes toward mukbang and para-social relationships.
„Based on the literature, this study builds a research model to investigate how a para-social relationship with mukbang content providers is related to the intention to watch mukbang.“
„The vital goal of this research is to explain the motivations behind mukbang watching“
Your research goes beyond the ambitions defined in the purpose.
3. If your goal is to analyse motives' impact on watching, why did you involve other factors?
The goal must be pursued incrementally throughout all the paper.
4. There are many studies, which analyses consumers’ engagement to consume content. Video watching is the lowest stage of consumer engagement behavior.
Some recommendations
Zailskaite-Jakste, L., & Minelgaite, I. (2021). Consumer engagement behavior perspective in social media: Mediating role and impact on brand equity. Journal of Eastern European and Central Asian Research (JEECAR), 8(2), 160-170.
5. “As a result of referring to these previous studies, research focused on factors impacting motivation to watch mukbang content is rare.“ Please provide the citation.
6. Try to make your sentences as simple as possible.
7. You have to answer to yourself: do you analyse intentions (intentions to watch content) or consumer engagement behavior (content watching)? Research phases 1 and 2 must be related to each other.
8. What is the difference between the motives that lead toward intentions to watch content and content watching (without intentions)?
9. How did you identify that motives (phase 1) make an impact on attitude and para-social relationships? "The results of the qualitative research into the motivation for watching mukbang..."
10. Maybe attitude and para-social relationships can be as moderating or mediating factors.
11. How many respondents participated in Quantitative research?
12. The survey was organized online or offline?
13. Have you tested the overall model fit?
14. It will be better to compare which variables make a stronger impact (weight) on the dependent variables and after this to make recommendations.
15. For which part of the population the conclusions you can apply?
Author Response
I would like to express gratitude to Reviewer 3 for the valuable review. After revision, I realized that I are indebted to the reviewer for every improvement made in the paper. The comments from Reviewer 3 and the corresponding replies/revisions are attached in the docu file.

Reviewer 4 Report
The paper tries to apply the SOR model to Mukbang content behavior. The authors also present a mixed-method study. However, theory development needs to be strengthened because the proposed methodology also is very simple to explain or evaluate this issue. Therefore, it will make it hard to find a new concept idea.
In the first point, the attitude behaviors, intention factors, and SOR model generally are derived and come from many theories including many factors which many previous papers have presented. They should be analyzed and synthesized by methods of step by step “How did you use and select such theory? Which are points of strong and weak?”.
For the second point, qualitative research is used for identifying the input factors to the SOR model. It is hard to accept for the reliability of a proposed model because the qualitative study will focus on narrow samples that it does not represent all samples. Therefore, it should design better the conceptual model with supporting the theory.
From above reasons, it will help your proposed model to prove and validate its reliability.
Other comments:
1. The paper should compare the affecting factors of the model between the previous research and your proposed model.
2. The ethics of the paper should be approved and consented from an institutional review board (IRB).
3. The grammar should be more improved for English writing.
Author Response
I would like to express gratitude to Reviewer 4 for the valuable review. After revision, I realized that I are indebted to the reviewer for every improvement made in the paper. The comments from Reviewer 4 and the corresponding replies/revisions are attached in the docu file.

Reviewer 5 Report
The work done by the researcher(s) is of great interest to the study area. The research phenomenon is properly worked out, the author(s) details each of the points appropriately. Each of the steps and the support given to the decisions taken is clear.
I believe that the slight adjustment to be made to the text revolves around two points:
1. Key terms are set out at the beginning of the text and should be clarified to the reader as parasocial. While they are explained below, this does not preclude clarification from the outset.
2. The analysis must be accompanied by evidence of the result, which makes it easier for the reader not to lose the argument.
Author Response
I would like to express gratitude to Reviewer 5 for the valuable review. After revision, I realized that I are indebted to the reviewer for every improvement made in the paper. The comments from Reviewer 5 and the corresponding replies/revisions are attached in the docu file.

Round 2
Reviewer 4 Report
The paper has been revised and reflected almost in the comments and suggestions. However, the output of purchase intention is purchase behavior, a fact that can be derived from many theories. Therefore, it should be discussed and compared more with previous research in 2. Literature Review and 7.1. Theoretical Contributions: "How is the proposed model that is the same and different in terms of based concept structure?"; because many theories can build up the purchase intention or purchase behavior e.g.
1. A retentive consumer behavior assessment model of the online purchase decision-making process https://doi.org/10.1016/j.heliyon.2021.e08169
2. A Holistic Perspective Model of Plenary Online Consumer Behaviors for Sustainable Guidelines of the Electronic Business Platforms. https://doi.org/10.3390/su14106131
Author Response
Reviewer: 4
I would like to express gratitude to Reviewer 4 for the great review in round 2. The comments from Reviewer 4 and the corresponding replies/revisions are as follows:
- The paper has been revised and reflected almost in the comments and suggestions. However, the output of purchase intention is purchase behavior, a fact that can be derived from many theories. Therefore, it should be discussed and compared more with previous research in 2. Literature Review and 7.1. Theoretical Contributions: "How is the proposed model that is the same and different in terms of based concept structure?"; because many theories can build up the purchase intention or purchase behavior e.g.
- A retentive consumer behavior assessment model of the online purchase decision-making process https://doi.org/10.1016/j.heliyon.2021.e08169
- A Holistic Perspective Model of Plenary Online Consumer Behaviors for Sustainable Guidelines of the Electronic Business Platforms. https://doi.org/10.3390/su14106131
Thank you for comments. In this study, we inserted some parts as follow. This makes our research enhance quality of research. So many thanks again!
|
1 |
Location |
2.5. Purchase Intention and Purchase Behavior |
|
From |
- |
|
|
To |
2.5. Purchase Intention and Purchase Behavior [45] tried to confirm relationship with consistent online platforms affecting purchase and re-purchase behaviors. This study confirmed the relationship between purchase intention and purchasing behavior. [46] showed the consuming behaviors integrating Push-Pull Mooring (PPM) model and the three stages of consuming behaviors in online environment. This study confirmed the relationship between pre-purchase and purchase. [47] investigated a model to test the effects of usability, functionality, and sociability on the purchase decision-making process. As such, the result that purchase intention has a significant effect on purchase behavior has been proven in several literature. Therefore, this research aims to investigate purchasing behavior by measuring purchase intention. |
|
1 |
Location |
7.1. Theoretical Contributions |
|
From |
- |
|
|
To |
Furthermore, this study is valuable since this research provides an academic clue to consumers' purchasing behavior and post-purchase behavior by illuminating watching contents intention and purchasing intention. The watching contents intention and purchasing intention in this study correspond to the stage of online purchasing decision-making process in [45]’s research. According to the [45], intention recognition enhances purchase behavior, eventually leading to recommend and re-purchase. Therefore, individuals with high viewing intention and purchasing intention may actually purchase services or products. Scholars can empirically analyze actual purchase behavior and post-purchase behavior in the context of this study. In addition, the watching intention and purchase intention in this paper correspond to the pre-purchase process among the consumer behavior stages of [46]. [46] validated that pre-purchase improves purchase, resulting in post-purchase. To sum up, users with high intention to watch contents and purchase will actually show purchasing behavior and perform post-purchase actions. |

Round 3
Reviewer 4 Report
The revised manuscript has been improved for responding to the comments.